# BOOSTING MEMBERSHIP INFERENCE ATTACKS WITH UPSTREAM MODIFICATION

## ABSTRACT

Membership Inference Attacks (MIAs) can be used by model owners to identify privacy leakage of specific points in their machine learning models. In this setting, the model owner (who is playing the role of the attacker) has perfect knowledge of the training data but a limited computational budget. However, current MIAs have limited effectiveness in this scenario: 1) They perform poorly against the most vulnerable points 2) They require training too many models. To overcome this weakness, we modify two limitations, in the initial/upstream stages of the MIA framework, namely sampling bias (i.e., too many points dropped during sampling) and attack aggregation (i.e., average attack results over all the data points instead of only the most vulnerable ones). Our improvements carryover downstream and boost attack accuracy of existing MIAs by *increasing the TPR of existing attacks at incredibly low FPRs (as low as zero) while achieving a near-perfect AUC*. As a consequence, our modifications enable the practical and effective application of MIAs for identification of data-leakage in machine learning models.

## 1 INTRODUCTION

MIAs are attacks designed to quantify training data leakage by predicting whether or not a point was used for training a given model. The most popular and effective of these attacks are the ones based on the shadow model framework Salem et al. (2018); Zarifzadeh et al. (2024); Carlini et al. (2022); Watson et al. (2021); Long et al. (2020); Sablayrolles et al. (2019); Song & Mittal (2021); Jayaraman et al. (2020); Shokri et al. (2017); Yeom et al. (2018). A model owner might be incentivized to use these attacks to identify which specific points are leaked by their model, so that they can take preventative measures.

However, MIAs are not designed for this specific scenario. They are designed for settings where the attacker has limited knowledge of the training data, but a large computation budget (in order to train dozens or even hundreds of shadow models). In contrast, current MIA do not fully leverage the availability of the training data, and unavailability of a large computation budget. As a result, even the most powerful MIAs suffer from the same **critical weakness: low power at low false positive rates (FPRs)**. This is an important limitation because, as pointed out by recent works, an attack is considered successful only if it *reliably* violates the privacy of data points without mistakes Carlini et al. (2022); Stadler et al. (2022); Aerni et al. (2024). In contrast, an attack that *unreliably* achieves a high aggregate success rate can be considered a failure. While a

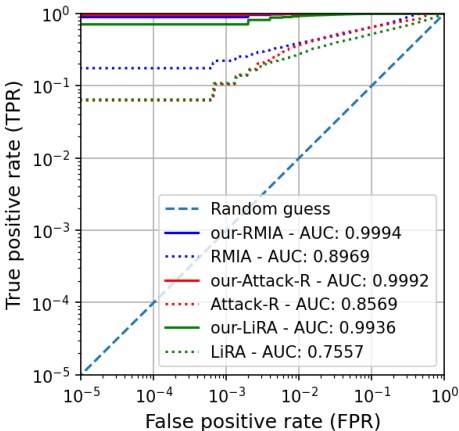

Figure 1: Original attack framework (dotted lines) vs our improved one (solid lines) for CIFAR-100 using only **one** model per query. We achieve a near perfect accuracy (both in terms of AUC and TPR at zero FPR).

large computational budget can alleviate this issue, this might not be possible for most model owners who do not have the budget to train multiple instances of their large commercial models.

Given the limited access to the target's training data, most attacks only focus on modifying the final stage of the pipeline (i.e., the hypothesis test) Carlini et al. (2022); Zarifzadeh et al. (2024); Bertran et al. (2024); Ye et al. (2022); Yeom et al. (2018). In our case, we assume the model owner wants to identify the leakage of specific points. As a result, we take a radically different approach. Instead of attending to the last stage of the framework, we focus our efforts upstream. We investigate and identify issues in the initial stages of the framework, and consequently, propose modifications. We find that our improvements carryover downstream, and are able to drastically increase accuracy of existing MIAs, at times, to near perfect (Figure 1). **As a result, our modifications significantly boost accuracy of *existing* MIAs and consistently achieves a high power across all FPRs (even as low as 0), using any computational budget.** In other words, we can reliably help a model owner identify which of their data points are being leaked under a low computational budget.

**Contributions:** We investigate the shadow model framework used for MIA attacks (Figure 2). In doing so, we uncover two limitations that upper bound the effectiveness of existing attacks.

The first limitation we encounter is at Step 2 of the framework (Figure 2), which we call *sampling bias* (Section 3.1). This is a result of dropping too many points during data set generation. In other words, when the drop rate is high, the attacks perform poorly for low FPRs. We find that lowering the drop rate can significantly improve attack success. For example, lowering the rate from 50%, commonly used in existing works Carlini et al. (2022); Zarifzadeh et al. (2024); Ye et al. (2022), to just 10% improves attack accuracy by $2\times$. One might be tempted to mitigate sampling bias by simply reducing the drop rate during data set generation. However, this will increases the number of data set partitions, and therefore, requires training more shadow models, ballooning the computational costs. For example, a 50% drop rate will result in $100\%/50\% = 2$ partitions, while a drop rate of 10% results in $100\%/10\% = 10$ partitions. In order to address this issue, we develop a robust partitioning method that uses a small drop rate without increasing the number of partitions. We discuss this method towards the end of this Section.

The second limitation we observe is at Step 4 of the framework (Figure 2), known as *attack aggregation* (Section 3.2). This is a well known issue in existing MIAs (not just shadow model ones), and is defined as the averaging of attack accuracy over all the points in the data set Carlini et al. (2022); Steinke & Ullman; Aerni et al. (2024). Instead, MIAs should focus only on the most vulnerable ones Carlini et al. (2022); Steinke & Ullman. Therefore, our contribution is to develop means to identify these vulnerable points, without having to run a MIA in the first place. We find that outlier detection methods perform exceptionally by improving TPR, by as much as, $11\times$ at zero FPR while achieving near perfect AUC, even for large datasets like CIFAR-100 and Tiny Imagenet.

Having described both limitations, we now combine our modifications to ensure maximal attack accuracy. We devise a sampling strategy that limits the number of partitions while simultaneously using a small drop rate. Specifically, instead of creating data partitions from the entire set, we *only* sample from the most vulnerable points (i.e., the outliers) (Section 4). After doing so, we perform an extensive evaluation of our modified framework across different attacks and datasets. We find that our modifications not only help achieve near perfect AUC but also *near perfect TPR at zero FPR*, using only a single shadow model. In fact, a single shadow model with our modified framework can out perform the original framework that uses 100 shadow models. To add to that, since our approach makes upstream modifications, it gives us the unique ability to easily integrate with, and boost the accuracy of, existing MIAs.

## 2 BACKGROUND AND RELATED WORK

### 2.1 THE MIA GAME

Before we discuss details of our modifications, lets take a moment to understand the MIA game. The goal of MIA is to determine whether a specific data point $x$ was used for training a given model $\theta$. MIA is defined by an indistinguishability game between a challenger and adversary (i.e., privacy auditor) Ye et al. (2022); Carlini et al. (2022); Zarifzadeh et al. (2024); Shokri et al. (2017); Homer et al. (2008); Sankararaman et al. (2009); Ye et al. (2022). The game models random experiments

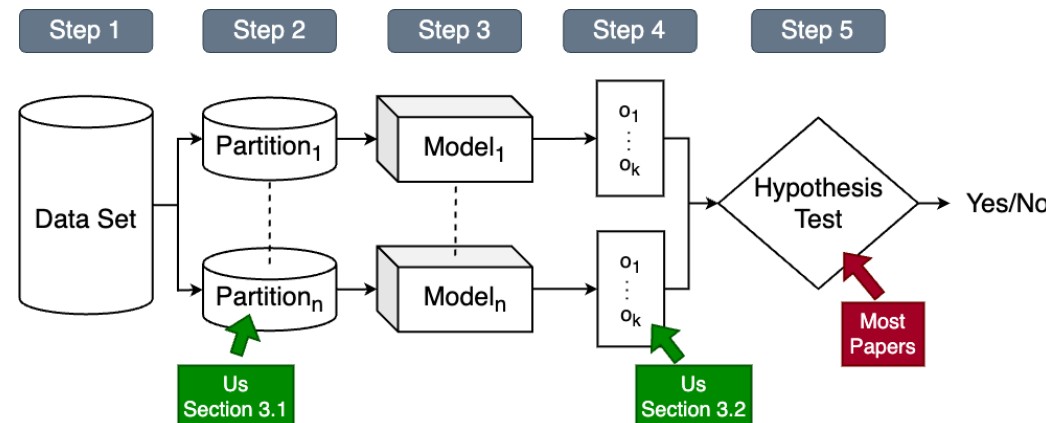

Figure 2: The figure shows the shadow model framework which is the cornerstone of the state-of-the-art MIAs. The framework comprises five steps. While most papers propose changes to the final step of the pipeline, we modify the upstream steps. In doing so, we can achieve significantly higher attack success over existing works.

related to two scenarios; $H_{in}$: the model $\theta$ was trained on $x$; and $H_{out}$: $x$ was not in $\theta$'s training set (the null hypothesis). The adversary has to decide whether or not point $x$ was present in the trained model $\theta$, using his background knowledge about the training algorithm and data distribution.

**Definition 2.1 (Membership Inference Game).** Let $\pi$ be the data distribution, and let $\mathcal{A}$ be the training algorithm.

1. The challenger samples a training dataset $S \sim \pi$, and trains a model $\theta \sim \mathcal{A}(S)$.

2. The challenger flips a fair coin $b$. If $b = 1$, it randomly samples a data point $x$ from $S$. Otherwise, it samples $x \sim \pi$, such that $x \notin S$. The challenger sends the target model $\theta$ and the target data point $x$ to the adversary.

3. The adversary, having access to the distribution over the population data $\pi$, computes $Score_{MIA}(x; \theta)$ and uses it to output a membership prediction bit $\hat{b} \leftarrow MIA(x; \theta)$.

A membership inference attack assigns a membership score $Score_{MIA}(x; \theta)$ to every pair of $(x, \theta)$, and performs the hypothesis testing by outputting a membership bit through comparing the score with a threshold $\beta$:

$$MIA(x; \theta) = I_{Score_{MIA}(x;\theta) \geq \beta} \tag{1}$$

The adversary's power (true positive rate) and error (false positive rate) are quantified over numerous repetitions of the MIA game experiment. The threshold $\beta$ controls the false-positive error the adversary is willing to tolerate Sankararaman et al. (2009); Ye et al. (2022); Bertran et al. (2024); Carlini et al. (2022).

The $Score_{MIA}(x; \theta)$ and the test equation (1) are designed to maximize the MIA test performance as its power (True Positive Rate or TPR) for any False Positive Rate (FPR). The (lower-bound for the) leakage of the ML algorithm is defined as the power-error trade-off curve (the ROC curve), which is derived from the outcome of the game experiments across all values of $\beta$. We primarily compare attacks based on their TPR-FPR curves but also analyze their computational efficiency and their stability (i.e., how much their power changes when we vary the data distribution and attacker's computational budget).

## 2.2 SHADOW MODEL FRAMEWORK

The most effective and popular means of playing the MIA game is via the use of the shadow model framework Salem et al. (2018); Zarifzadeh et al. (2024); Carlini et al. (2022); Watson et al. (2021);

Long et al. (2020); Sablayrolles et al. (2019); Song & Mittal (2021); Jayaraman et al. (2020); Shokri et al. (2017); Yeom et al. (2018). It involves training local (or shadow) models to mimic the target model, using auxiliary data from the same distribution as the target. In doing so, an attacker can understand how the target model might behave when a data point is present or absent, enabling accurate membership inferences.

Specifically, this framework, shown in Figure 2, involves the following steps:

- **Step 1 - Collect Data Set:** The first step is to collect an auxiliary data set from the same distribution as the target model's training data. This data serves as a substitute for the actual training data of the target model, which the attacker typically does not have access to. Auxiliary data can be obtained from public data sets, augmentation, or similar sources.

- **Step 2 - Generate Data Partitions:** Split the data into multiple subsets or partitions. These are used for training shadow models to simulate different scenarios of what the target model might have been trained on. The more the partitions, the more the shadow models. In the *ideal* world, subset generation should only remove a single point from the dataset (or run a leave-one-out test). This will provide a precise understanding of how the model behaves when a single point is removed. However, this setup is computationally infeasible for even small datasets and models as it requires training as many models as points in the set. To overcome this issue, data set generation partitions the data by dropping a specific percentage of random points. As we discuss in the next Section, current MIAs drop too many points during partitioning, leading to a sampling bias, and consequently, poor MIA results. Therefore, we modify this step to improve MIA accuracy.

- **Step 3 - Train Shadow Models:** Train each shadow model on one of the partitions generated in the previous step. To improve the accuracy of the MIA, authors train multiple shadow models *per query*, i.e., the number of models with a given point present/absent from the training data. The total number of shadow models that the attacker will need to train is $partitions \times models\_per\_query$. The number of models per query can range from a few dozen Zarifzadeh et al. (2024) to a few hundred Carlini et al. (2022). Fortunately, we find that our proposed modifications can out perform existing the original framework while requiring only a single model per query.

- **Step 4 - Get Shadow Model Predictions:** For each data point in the dataset, collect the models' predictions, confidence scores, loss values, or gradient information, etc. These are then used by the final step to predict whether or not a given point was present in the training data. As we discuss in the next Section, current MIAs collect predictions over all the points in the set. Instead, we modify this step to focus on only the most vulnerable points, thereby improving MIA accuracy.

- **Step 5 - Hypothesis Test:** Finally, the attacker uses statistical tests to over model predictions to predict whether a data point was a present or absent during training. Most MIA papers focus on this step Salem et al. (2018); Zarifzadeh et al. (2024); Carlini et al. (2022); Watson et al. (2021); Long et al. (2020); Sablayrolles et al. (2019); Song & Mittal (2021); Jayaraman et al. (2020); Shokri et al. (2017); Yeom et al. (2018), modifying the statistical tests to scale up attack accuracy without having to train additional models.

## 2.3 THREAT MODEL

We assume a setting where the model owner wants to determine which points are most vulnerable to privacy leakage. As a result, the model owner plays the role of the attacker and uses current state-of-the-art MIAs for this purpose. We make two assumptions:

1. We assume that the attacker (in this case the model owner) has complete access to the model was trained on (i.e., large overlap between shadow and target model). This is not a realistic assumption for outside attackers as they are unlikely to have perfect knowledge of the training data. However, in our case, since the model owner is also "attacker", this is a perfectly acceptable assumption Ye et al. (2022).

2. We assume that the model owner has a limited computation budget. This is means that they can do not have the compute to train dozens or even hundreds of models, that are

often required by current MIAs. This is also an acceptable assumption as training multiple instances of large commercial models might not be possible for most entities.

Though effective in outside attacker scenario, current MIA have limited use in this setting. They either perform poorly on the most vulnerable points, or require training too many models. Our goal is to help bridge this gap by proposing modifications to existing MIAs that can help boost performance in this scenario.

## 3 LIMITATIONS AND MODIFICATIONS TO THE EXISTING FRAMEWORK:

Having described the framework, we can now explore its limitations, and recommend modifications. While most MIAs focus on the last step of the pipeline (Step 5), we focus upstream (Steps 2 and 4). We find that our improvements are carried downstream thereby increasing overall MIA accuracy.

### 3.1 SAMPLING BIAS

**Limitation:** The first limitation we identify is at Step 2 of the MIA framework (Figure 2). This step generates data partitions by dropping random points from the dataset. In most attacks, this drop rate is around 50% Carlini et al. (2022); Zarifzadeh et al. (2024); Ye et al. (2022). This means, for a set of 50,000 points (e.g., CIFAR-10), the MIA algorithm will drop at least 25,000 to create a single partition. This leads to significant issue: The more points we drop, the further we stray from the idealized leave-one-out experiment. As consequence, dropping *too* points can remove entire sub-populations, leading to a sampling bias Abdullah (2024). If the partition does not contain points from the small sub-population, the corresponding shadow models trained in the next step stage (Step 3) will produce biased outputs (Step 4), resulting in poor attack accuracy (Step 5). As a result, modifying this step has the potential to improve all the downstream tasks.

**Modification:** We hypothesize that these biased predictions will result in a low power at a low FPR. Therefore, in order to overcome the issue, we need to drop fewer points during sampling.

**Setup:** To evaluate this hypothesis, we follow the training parameters laid out in previous work Carlini et al. (2022); Zarifzadeh et al. (2024). We use different drop rates (50%, 40%, 30%, 20%, 10%) and record the corresponding change in attack success. Due to the computationally intensive nature of this experiment (repeating it for different drop rates), we run the evaluation only over CIFAR-10, a benchmark dataset commonly used for MIA Carlini et al. (2022); Zarifzadeh et al. (2024); Aerni et al. (2024). While we do use more complex datasets (CIFAR-100 and Imagenet) in the rest of the paper, our goal for this experiment is to reveal limitations in the existing framework. Therefore, using just a single dataset is sufficient, at least for this experiment Aerni et al. (2024). Next, we train RESNET-18 models on each of the partitions. We use a batch size of 512, with a triangular learning rate of 0.4, and weight decay of $5e^4$. We use the FFCV library Leclerc et al. (2023) to improve the training speed. Having trained the models, we now run the RMIA Zarifzadeh et al. (2024), Attack-R Ye et al. (2022), and LiRA Carlini et al. (2022) attack, which is the most effective one in the current literature. We use attack implementations available in the Privacy Meter repository Kumar & Shokri (2020). We use the standard metrics, notably FPR versus TPR curve, and the Area Under the Curve (AUC), for analyzing attack success. Unless otherwise specified, we use these models and metrics for all the experiments of this work.

**Results:** Figure 3 shows the results of running the RMIA attack across different drop rates, averaged over 10 target models. We can see as the drop rate decreases, the attack success rate increases. Specifically, decreasing the drop rate from 50% to 10%, improves TPR at zero FPR by $2\times$ (0.73 to 1.41). This trend is consistent across the remaining two attacks as well (Figure A.1). This clearly shows that dropping too many points does lead to a sampling bias, thereby lowering attack success.

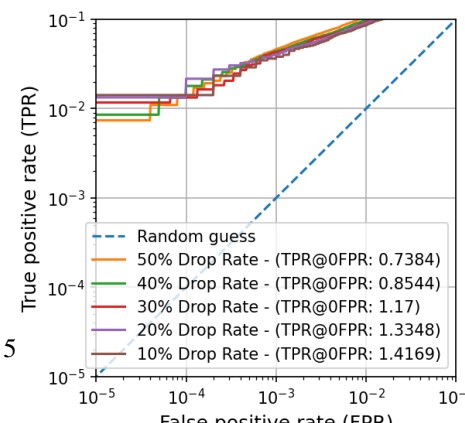

However, simply using a lower drop rate during data set partitioning would be flawed. As we discussion in Section 1. This is precisely why MIAs do not execute the ideal leave-one-out approach (i.e., create a partition by dropping only a single point), even though it yields the most accurate results. In the next section, we propose a robust sampling method to overcome this challenge . For now, it is important to remember that the lower the drop rate, the better the MIA results.

## 3.2 Attack Aggregation:

**Limitation:** The next limitation we identify is at Step 4 of the framework. Here, the MIA extracts the output statistics (such as output confidence, logit value, etc) from the shadow models. Usually, MIAs generate the output for *every* point in the data set. However, recent work has shown that privacy is not an average case metric and an ideal MIA should only focus on the most vulnerable points Carlini et al. (2022); Steinke & Ullman; Aerni et al. (2024). While the idea itself is well known, what is not clear is *how* to identify these vulnerable points in the first place.

There are a few simple (yet flawed) approaches one could take here. For example, recent work has proposed using of artificial canaries to mimic vulnerable data Aerni et al. (2024). However, this approach is unsound for three reasons reasons. First, artificial points do not represent the actual training distribution. Therefore, findings from artificial studying artificial points might not even apply to real data Abdullah et al. (2023). Second, MIAs are used for auditing model privacy. Model owners are more concerned with privacy exposure of *existing* training points instead of the artificial ones. Lastly, training models on artificial data can hurt generalization, thereby limiting model utility Arpit et al. (2017); Raghunathan et al. (2019).

Another simple approach would be to run the MIA to extract the most vulnerable points. However, this too has a number of serious limitations. First, under the current framework, MIAs require training many additional models, making it infeasible for any real-world use case. Second, current MIAs themselves have a number of limitations, which is the main point of this work. Therefore, using MIA for outlier detection is flawed[1].

**Modification:** To that end, we hypothesize that outliers constitute the most vulnerable points to membership inference. Therefore, we can significantly improve MIA success by first identifying and then attacking only the outlier points.

**MIA and Outliers vs Inliers:** Before we evaluate our hypothesis, let's first discuss the relationship between outliers, inliers, and membership inference. An outlier is a data point that significantly deviates from the other observations in a dataset. Outliers can include: rare points (e.g., a single pink cat in a dataset of white cats), malformed inputs (e.g., a blurry or garbled cat), or even mislabeled points (e.g., a cat mislabeled as an `ORANGE`), etc. Outliers have few (if any) overlapping features with the remaining points in the label set (i.e., rare, malformed, and mislabeled inputs do not look like the other images in the label set). As a result, the features learned from the rest of the label distribution are not useful for classifying these points Feldman (2020); Abdullah et al. (2023); Abdullah (2024). Therefore, the model ability to classify these points correctly is largely subject to their inclusion in the training data. In other words, the shadow model output for outliers will differ based on whether these points were present in the training data. For example, the cat mislabeled as an `ORANGE` will likely be classified as an orange when it is present in the shadow model data but might be classified as `CAT` when absent. As a result, membership inference of outliers is easiest compared to other points in the rest of the dataset Carlini et al. (2022; 2023; 2021); Somepalli et al. (2023); Ye et al. (2022).

In contrast, inlier points are the least vulnerable to membership inference attacks. They are classified correctly even when absent from the dataset. This is because these points have overlapping features with the rest of the distribution. Features learned from remaining points in the dataset are sufficient to

---

[1]For completeness, we still do evaluate this approach in the Appendix A.2.

Table 1: Performance of RMIA over different data sets using various outlier detection methods using only 5,000 points. We use only **one** shadow model per query and average the results over 10 random models. ApB (in **bold**) outperforms all other methods across all data sets. It achieves either very high or near perfect AUC. Incredibly, it also achieves a near perfect TPR at 0% FPR against tiny-Imagenet, a large and complex dataset.

| | Baseline | TPR@FPR | | SPP | TPR@FPR | | MD | TPR@FPR | | ApB | TPR@FPR | |
|---|---|---|---|---|---|---|---|---|---|---|---|---|
| | AUC | 0% | 1% | AUC | 0% | 1% | AUC | 0% | 1% | AUC | 0% | 1% |
| CIFAR-10 | 65.87 | 2.99 | 8.98 | 83.3 | 7.1 | 22.04 | 80.08 | 7.25 | 18.96 | **95.81** | **11.81** | **45.29** |
| CIFAR-100 | 89.71 | 17.1 | 38.75 | 97.1 | 32.48 | 61.63 | 91.15 | 16.45 | 41.08 | **99.94** | **74.46** | **98.87** |
| Tiny-Imagenet | 94.71 | 9.35 | 47.71 | 99.63 | 42.43 | 89.76 | 95.1 | 14.2 | 47.99 | **100.0** | **99.83** | **100.0** |

correctly classify the missing inliers. As a result, the shadow model output will not vary for inliers, thereby making membership inference harder. Therefore, our goal differentiate the (vulnerable) outliers from the (invulnerable) inliers, in order to be able to improve MIA accuracy.

**Setup:** We now evaluate our hypothesis. We execute our evaluation on CIFAR-10 and CIFAR-100 datasets. To further validate our hypothesis, we also run our evaluation on Tiny-Imagenet, which consists of 100,000 images across 200 classes. We use the same experimental setup outlined earlier.

We compare different outlier detection methods for this purpose, one from each of three broad families: probability-based, distance-based, and memorization-based. Specifically, use SPP (Softmax Prediction Probability) Hendrycks & Gimpel (2016), MD (Mahanobis Distance) McLachlan (1999), and ApB (Accuracy per Batch) Abdullah (2024) (we provide details of each of these methods in the Appendix). We train a single model on the full dataset to identify the outliers according to each of these methods. We pick the 1250, 2500, and 5000 points with the highest outlier score i.e., ones that are most likely to be outliers. We also sample an equal number of random points from the dataset, as a baseline, to compare the outlier results. Finally, we run the RMIA attacks, the strongest one in current literature, and compare our findings.

**Results:** Table 1 and Table 2 shows the experimental results. We can make three observations:

- *Every outlier detection method outperforms the baseline by significant margins:* We can see this trend hold across every data set in Table 1. The AUC increases by as much as 30 percentage points (CIFAR-10) and the TPR at FPR increases by as much as $11\times$ (Tiny-Imagenet). This demonstrates that outliers provide a good means of identifying points vulnerable to MIAs and should be used for evaluating the privacy metric.

- *ApB is the most effective outlier detection method:* Of the three outlier detection methods we evaluate (SPP, MD, and ApB), ApB is most effective against every data set in Table 1. We can observe that ApB improves the AUC by as much as 11 percentage points in the case of CIFAR-10. Similarly, for CIFAR-100 and Tiny-Imagenet, ApB helps achieve a near perfect AUC, and even a near perfect TPR at zero FPR for Tiny-Imagenet.

- *The stronger the outlier, the better the MIA results:* Table 2 shows how the attack accuracy improves when we pick the top N most outlying points for the CIFAR-10 data set. In every case, we see that the outliers out perform the baseline. For example, top 1250 outliers improve AUC by at least 20 percentage points. Additionally, we can see the difference between the outlier method also growing, with ApB out performing all other methods. Specifically, at 1250 points, ApB helps achieve near perfect AUC and improving TPR at FPR by $13\times$ over the baseline.

Collectively, these results confirm our hypothesis that outliers are most vulnerable to MIAs. Therefore, executing MIA on just the outliers can significantly improve the attack success rate. And of all the methods evaluated, the ApB score is by far the most effective.

Table 2: RMIA attack performance on CIFAR-10 across the top outlier points (first column) based on outlier detection methods. The top points represents the strongest outliers according to each of the detection methods. We use only **one** shadow model per query and average the results over 10 random targets. Similar to Table 1, ApB (in **bold**) outperforms all other methods, improving TPR (over the baseline) by *at least* 3×, respectively. Therefore, further strengthening the case for modifying Step 4 (Figure 2) to only use the outliers.

| Top Outliers | Baseline | | | SPP | | | MD | | | ApB | | |
|---|---|---|---|---|---|---|---|---|---|---|---|---|
| | AUC | TPR@FPR | | AUC | TPR@FPR | | AUC | TPR@FPR | | AUC | TPR@FPR | |
| | | 0% | 1% | | 0% | 1% | | 0% | 1% | | 0% | 1% |
| 1250 | 66.27 | 4.7 | 9.52 | 88.76 | 16.1 | 29.89 | 86.07 | 16.24 | 26.95 | **99.49** | **55.29** | **81.81** |
| 2500 | 65.97 | 3.49 | 7.81 | 86.48 | 10.88 | 24.54 | 83.51 | 10.37 | 22.08 | **98.47** | **24.78** | **68.68** |
| 5000 | 65.87 | 2.99 | 8.98 | 83.3 | 7.1 | 22.04 | 80.08 | 7.25 | 18.96 | **95.81** | **11.81** | **45.29** |

## 4 COMBINING MODIFICATION

In the last section, we explored the limitations of the existing framework and proposed two modifications to mitigate them. The first involves reducing the drop rate during dataset partitioning (Step 2 Figure 2). The second involves running the MIA against only the outliers, instead of the entire dataset (Step 4 Figure 2). In this section, we combine these modifications to achieve maximal MIA performance and then compare the results against the original framework.

### 4.1 COMBINING MODIFICATIONS

The simple (yet flawed) strategy to combine the modifications would be to first generate partitions using a small drop rate, train the models, and get the output statistics for only the vulnerable points. For example, use a drop rate of 5% to produce $100/5 = 20$ partitions. Train one shadow model per query on each partition, resulting in 20 models. Subsequently, use each models output statistics to execute the MIA. However, this strategy has a clear limitation as it requires training 20 models.

We propose to fix this issue by creating partitions from *only* vulnerable points. Specifically, 1) we train a single model on all points in the dataset of size $k$ and identify the vulnerable ones (i.e., outliers). 2) We split the dataset into two sets: vulnerable $v$ and non-vulnerable points $v'$, where $k = v + v'$. 3) During dataset generation, we sample from *only* the vulnerable points by randomly dropping a fraction of them. If the drop rate is $d$, then each partition will be of size $(1 - d) \cdot v$. 4) We append the non-vulnerable set to the vulnerable partitions. Each data shard will contain all the non-vulnerable points but only a sub-sample of the vulnerable ones. As a result, each partition will be of the size $v' + (1 - d) \cdot v$.

This setup will overcome the aforementioned limitations. If we select the 5,000 vulnerable points, a drop rate is 50% over them will result in $100/50 = 2$ partitions of 2,500 points each. This is 5% of the total dataset for CIFAR-10/CIFAR-100. This means we can use a drop rate of 5% over the total data but only have to train two models, instead of 20.

### 4.2 EVALUATION

**Setup:** We use the same setup as described in the earlier sections. Due to the overwhelming attack efficacy of the ApB detection method, we employ it for the rest of this section. For a fair comparison, we select 5,000 random points for the baseline and compare the results against 5,000 ApB outliers.

**How effective are our modifications over the original framework?**

Figures 1 and 4 show the ROC curves for the three datasets across three different attacks when using a single shadow model per query. Table 3 shows the specific TPR at FPR values in addition to the AUC scores for each experiment. We can see that our modified attacks (solid line) strictly outperform the original ones (dotted line) in every single case. Additionally, the original framework has single digit TPR across all attacks at zero FPR. In stark contrast, our modifications lead to significantly better results. In fact, in the case of CIFAR-100 and Tiny-Imagenet, our modified

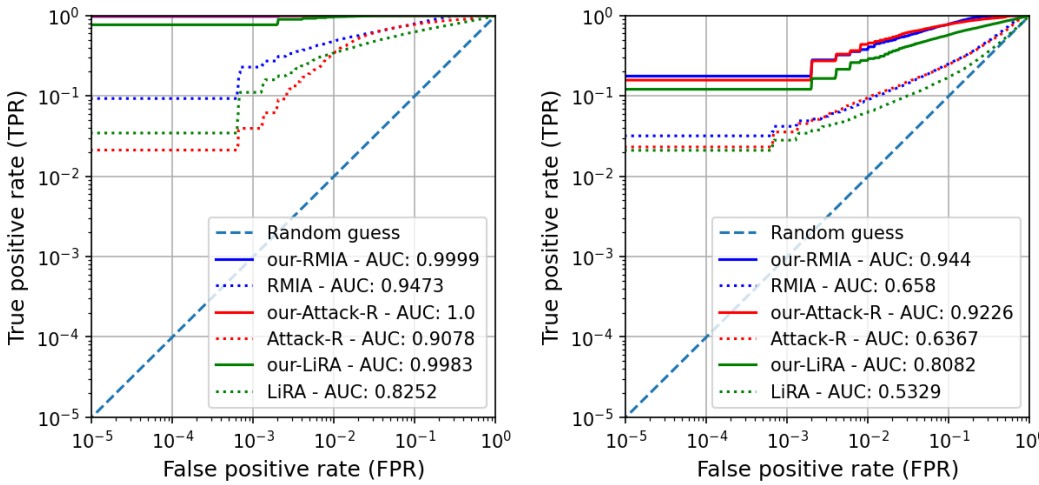

Figure 4: Original attack framework (dotted lines) vs our improved one (solid lines) using **one** shadow model per query. Left: Tiny-Imagenet. Right: CIFAR-10. Our modifications outperform the original framework by significant margins.

Table 3: The original attack compared against our combined modifications. Our modifications outperform the corresponding unmodified pipeline in *every* case, even when the original uses 100 models.

| Models Per Query | Attack | CIFAR-10 | | | CIFAR-100 | | | Tiny Imagenet | | |
|---|---|---|---|---|---|---|---|---|---|---|
| | | AUC | TPR@FPR | | AUC | TPR@FPR | | AUC | TPR@FPR | |
| | | | 0% | 1% | | 0% | 1% | | 0% | 1% |
| 1 | LiRA | 53.22 | 2.12 | 6.29 | 75.6 | 6.34 | 27.48 | 82.56 | 3.82 | 34.28 |
| 100 | LiRA | 55.03 | 2.72 | 8.44 | 80.68 | 19.49 | 43.55 | 88.19 | 15.87 | 53.0 |
| **1** | **our LiRA** | **80.84** | **12.49** | **28.68** | **99.37** | **70.0** | **92.48** | **99.84** | **80.22** | **96.2** |
| 1 | Attack-R | 63.72 | 2.14 | 9.48 | 85.7 | 6.73 | 36.71 | 90.79 | 2.3 | 33.63 |
| 100 | Attack-R | 64.37 | 4.68 | 10.85 | 85.17 | 30.14 | 46.95 | 90.72 | 37.36 | 57.35 |
| **1** | **our Attack-R** | **92.25** | **15.31** | **44.43** | **99.92** | **95.33** | **99.12** | **100.0** | **99.37** | **99.99** |
| 1 | RMIA | 65.87 | 2.99 | 8.98 | 89.71 | 17.1 | 38.75 | 94.71 | 9.35 | 47.71 |
| 100 | RMIA | 68.3 | 5.46 | 13.37 | 92.26 | 36.12 | 51.31 | 96.2 | 38.94 | 60.76 |
| **1** | **our RMIA** | **94.39** | **18.63** | **38.71** | **99.94** | **88.75** | **98.7** | **99.99** | **97.54** | **99.92** |

ATTACK-R achieves not only near-perfect AUC but also a near-perfect TPR at zero FPR. This means our modified MIA can successfully exploit 5% of the total data in Tiny-Imagenet, and 10% from CIFAR-100, without making any mistakes.

**Does the framework boost the accuracy of each MIA?**

Our modifications are made in the earlier stages of the pipeline. And the results show that they carry over downstream and improve the accuracy of each MIA we evaluate. In fact, relative attack accuracies remain consistent after modifications i.e., attacks that were strongest before modifications are also the strongest after modifications. For example, RMIA is the most accurate, followed by Attack-R and LiRA since it has the highest AUC. After our modifications, the attack accuracy order remains unchanged, with RMIA having the highest AUC (in all but one dataset), followed by Attack-R and LiRA. This shows that upstream improvements carry into the downstream stages, improving the overall attack results.

**How does the attack perform against large datasets?**

We can see that the larger and more complex the dataset, the better the attack accuracy. For example, in Table 3, the AUC of CIFAR-100 and Tiny-Imagenet are near perfect, with the TPR at zero

FPR reaching near 100% on both our version of RMIA and Attack-R. In contrast, the attacks with the original framework achieve, at best, 9.35 TPR at zero FPR and AUC of 94.71 when using a single model per query. This means our modifications improve the TPR by $11\times$. Therefore, our modifications are particularly effective against large and more complex datasets.

**Why do our improvements perform better on large datasets than the smaller ones?**

One interesting observation to note is that attack accuracy for CIFAR-10 using our method is slightly lower than for CIFAR-100 and Tiny-Imagenet. A natural question that emerges is, why? Popular belief would indicate that larger datasets should be *harder* to attack Zarifzadeh et al. (2024); Bertran et al. (2024). Instead, our experiments demonstrate the contrary. Fortunately, a simple explanation exists. Recent work from the space of memorization has shown that models memorize more points from larger, more complex datasets (e.g., CIFAR-100, Tiny-Imagenet) Abdullah et al. (2023). However, when the dataset is small and simple (e.g., CIFAR-10), models can generalize better, resulting in less memorization. Since there is more memorization in large datasets, more of these points should be vulnerable to MIAs. Our results corroborate these findings by showing that the top outliers are easily detectable using MIAs.

**Is the cost of training an extra model for outlier detection worth the results?**

An astute reader will remember that our modification in Step 4 requires training one additional model on the full data to identify the outlier points. Having demonstrated the overwhelming efficacy of our attacks, a natural question that emerges is whether the cost is worth the improvement in attack success. Therefore, we compare the modified attack with just one shadow model per query (plus one extra model for outlier detection) against an original attack with 100 shadow models per query in Table 3. Even in this extreme case, where the original framework has two orders of magnitude more shadow models, our modified attack (with just one shadow model) wins by significant margins. In fact, the original framework fails to reach 100% accuracy across any metric. In contrast, our method achieves perfect AUC and TPR at zero FPR scores across multiple datasets.

**How can the model owner identify *all* the leaked points?**

While this paper only targets the top 5,000 for evaluation purposes, it is easy to see how a model owner can extend it to the rest of the dataset. Table 2 shows that the stronger a point's outlier score, the more vulnerable it is to membership inference. The attacker can use this leverage this knowledge to asses overall leakage: 1) Select the top N points, where N is less than dataset size, 2) partition the N points, 3) train the shadow models, and 4) run the MIA. Up to now, the attacker is doing exactly what we do in Section 4. However, in order to evaluate the privacy of the remaining points, the attacker can now select the next N points and repeat steps 1-4. The attacker can continue to do this until they start noticing a significant reduction in attack success. At this point, the attacker has reached the inlier points and may decide not to proceed further. This enables the attacker to use their computational resources efficiently: Spend computation on attacking only the outliers without wasting computation on inliers.

## 5    CONCLUSION

In our work, we demonstrate the effectiveness of modifying the initial stages of the shadow model framework to improve MIA accuracy. Despite advancements, current state-of-the-art methods do not fully leverage the complete availability present when a model owner wants to evaluate data leakage on a limited computation budget. To overcome this challenge, we address two critical issues in the initial stages of the MIA process: sampling bias and attack aggregation. By resolving them, we dramatically enhance the TPR of existing attacks, particularly in low false positive rate FPR, achieving perfect scores across multiple metrics. As a result, our modifications make MIAs far more practical and reliable for assessing privacy risks in machine learning models.

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

# A APPENDIX

## A.1 OUTLIER DETECTION METHODS

However, there are a number of outlier detection algorithms that we can evaluate. Generally speaking, outlier detection algorithms can be divided into three broad categories[2] (Exploring the Limits of Out-of-Distribution Detection):

- **Probability Based:** A standard method for detecting OOD inputs are ones based on model's output probabilityHendrycks & Gimpel (2016). The higher the label probability, the lower the point is an outlier. Despite being somewhat less effective compared to other methods, its ease of implementation and solid performance make it a valuable baseline approach.

- **Distance Based:** Here, the point embeddings are used to find ones furthest from the class center. The further the point, the higher the likelihood of it being an outlier. One of the most popular distance based methods is ones based on Mahanobis Distance ?. These use a Gaussian distribution fit to class-specific embeddings for OOD detection. Let $f(\mathbf{x})$ represent the embedding (for instance, the output from the penultimate layer before

---

[2]As a reminder to the readers, this is not an exhaustive list of outliers detection methods from the rich existing literature. Instead, we focus on a three *very* broad categories, and leave evaluation using other methods for future researchers.

the logits) of an input $\mathbf{x}$. A Gaussian distribution is fitted to the embeddings from the training data, calculating the per-class mean $\mu_c = \frac{1}{N_c} \sum_{i:y_i=c} f(\mathbf{x}_i)$ and a shared co-variance matrix $\Sigma = \frac{1}{N} \sum_{c=1}^{K} \sum_{i:y_i=c} (f(\mathbf{x}_i) - \mu_c)(f(\mathbf{x}_i) - \mu_c)^\top$. The Mahalanobis score (which is the negative of the distance) is then determined as: $score_{Maha}(\mathbf{x}) = -\min_c \left( \frac{1}{2} (f(\mathbf{x}) - \mu_c)^\top \Sigma^{-1} (f(\mathbf{x}) - \mu_c) \right)$.

- **Memorization Based:** These are based on the idea that points that are easiest to "memorize" are most likely to be outliers . Here, memorization score is the number of shadow model that correctly classify the points when it is present in the dataset, compared to when it is absent. However, memorization scores require training thousands of models and therefore it is computationally expensive even for small data sets, and intractable for large ones. To overcome this limitation, researchers have proposed a proxy for memorization scores known as Accuracy Per Batch (ApB) . Here, the user trains a single model on the full dataset and counts the number of times a point is classified correctly at the end of each batch. The lower the ApB, the harder it was for the model to learn the point, and higher the likelihood the point is an outlier.

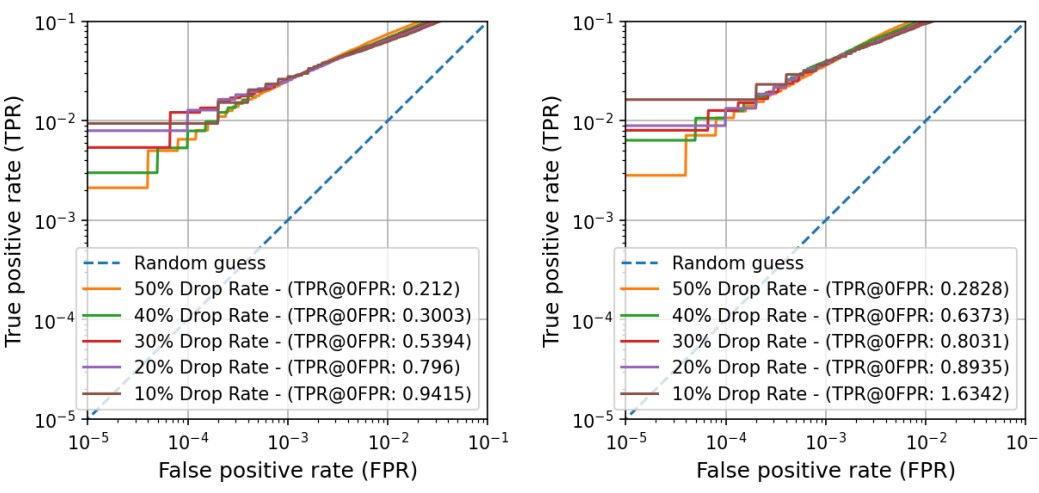

## A.2 MIA BASED OUTLIERS VIS APB

In Section 3.2, we evaluate with different outlier detection methods to see which one identifies the most vulnerable to MIA attacks. In this subsection, we evaluate one more technique: specifically, we use MIAs to extract outliers. As mentioned earlier in Section 3.2, outliers are most vulnerable to MIAs. Therefore, one way to identify outliers is to execute the attack and mark the inferred points as outliers.

To do so, we use RMIA, the most powerful of all the MIA attacks that we evaluate in this work. We use a drop out rate of 50% on the CIFAR-10 data set, resulting in two partitions, and therefore two shadow models. We use one shadow model per query. Next, we train a single target model to identify the points that were inferred correctly. We select the top 5,000 points and mark them as outliers. This is a total of three models (two shadow and one target). For attack evaluation, we run RMIA attack on 10 different target models and aggregate the attack success over only these 5,000 points. Next, we use the 5,000 outliers from ApB algorithm and aggregate the RMIA attack success over 10 target models.

Table 4 shows the results of our experiments. We can see that even though it required 3 models to extract the RMIA outliers, ApB outliers have a higher attack success. This means even using the most powerful MIA algorithm to extract the vulnerable points is not a viable alternative to existing outliers methods.

| | | CIFAR-10 | | |
| | Total Models | AUC | TPR@FPR | |
| | | | 0% | 1% |
| RMIA Outliers | 3 | 89.04 | 7.67 | 27.04 |
| **ApB Outliers** | **1** | **95.81** | **11.81** | **45.29** |

Table 4: Comparison between outliers extracted via RMIA against ApB.

