# OpenReview forum: "Boosting Membership Inference Attacks with Upstream Modification"
_ICLR.cc/2025/Conference — Submitted to ICLR 2025_

### Official Review · Reviewer_FBG3 · 2024-10-28

**Soundness:** 3
**Presentation:** 3
**Contribution:** 2
**Rating:** 5
**Confidence:** 4

**Summary:**

This paper investigates and proposes improvements to the membership inference attack (MIA) framework used to assess privacy risks in machine learning models. The authors identify two main limitations in current MIAs: sampling bias and attack aggregation. They show that a high data drop rate in sampling introduces bias and that averaging attack accuracy across all data points dilutes the impact of outliers, which are often more vulnerable to MIAs. Based on this finding, they propose creating data partitions not from the entire dataset but solely from the most vulnerable points selected through outlier detection methods. Extensive evaluations demonstrate that their approach improves the true positive rate (TPR) while reducing the false positive rate (FPR), thereby enhancing the effectiveness of MIAs in privacy assessment.

**Strengths:**

- This paper addresses limitations in MIA processes, which is crucial for improving privacy assessments as machine learning models continue to grow in complexity and application.
- This paper thoroughly evaluates challenges, such as sampling bias and attack aggregation, demonstrating that upstream modifications can yield significant improvements in MIA effectiveness across various datasets.
- The proposed approach, particularly the focus on outlier detection for data partitioning, leads to notable increases in true positive rates while maintaining low false positive rates, outperforming traditional MIA techniques.

**Weaknesses:**

- This paper lacks an in-depth exploration of the feasibility of the proposed improvement strategy. Although experiments indicate that as the drop rate decreases, the attack success rate increases, the paper does not clarify the relationship between a low drop rate and the selection of the most vulnerable samples. Furthermore, it does not clearly explain why MIAs should focus exclusively on the most vulnerable samples(outliers).
- While the paper’s modifications achieve improved performance at low false positive rates, a comparison with defense mechanisms (such as differential privacy) under similar FPR conditions would make the findings more actionable. Assessing whether the method still improves MIA accuracy when applied to models with built-in defenses could demonstrate its real-world applicability for assessing privacy risks.
- The experimental setup lacks a detailed description of the target model, evaluation metrics, and the baseline method (RMIA and LiRA) implementation. More experimental details are also needed. The paper provides only a brief description of model selection and parameter settings on CIFAR-10 in Section 3.1, without detailing the experimental setup on other datasets.
- Code is not released for reproducibility.

**Questions:**

Q1: The paper proposes a partitioning strategy based on outlier detection. Does this approach impact membership inference attack results for inliers?

Q2: Does the proposed improvement strategy rely entirely on the accuracy of the chosen outlier detection method, ApB? How should the approach handle data points that are missed or incorrectly identified by the detection method?

Q3: One motivation for the modified partitioning strategy is to reduce the computational budget, but it remains unclear if the additional cost for outlier detection is acceptable. Quantitative assessments of computational cost and scalability analysis with increasing model complexity would offer practical insights for real-world application.

---

> ### Author Response · Authors · 2024-11-24
>
> > Furthermore, it does not clearly explain why MIAs should focus exclusively on the most vulnerable samples(outliers).
>
> We thank the reviewer for this point. By studying the most vulnerable points, we are relying on the notion of  *individual privacy*. This notion is accepted within the community and has been echoed by a number of papers [5,6,7,8,9]. The most important of these works is by Aerni et al [5], who state “privacy is not an average-case metric! [54]”. They lay out several reasons for this:
>
> 1. “A blatantly non-private defense that fully leaks one training sample passes existing evaluations.” This demonstrates that even improved evaluation metrics fail to capture the true leakage in a model.
>
> 2. “Existing metrics can be arbitrarily “diluted” by adding new members for which a defense preserves privacy…” giving the illusion of privacy.
>
> 3. “this metric (and prior ones) [i.e., average case metrics] fail to properly capture *individual privacy.*”. Specifically, “If a model violates the privacy of an individual, that individual likely does not care whether the model also leaks 0.1% or 10% of the remaining samples; the individual cares about the fact that an attacker can confidently recover their data.”
>
> 4. Aggregating over the most vulnerable samples (instead of the entire dataset) helps “membership inference attacks to lower-bound the DP guarantees of an algorithm [27, 43, 53, 60].”
>
> These are some reasons why an individual notion of privacy is more important than average-case statistics. In this paper, we are not introducing a new notion, but instead, relying on an existing one that is accepted within the community. We would be happy to clarify this in the paper.
>
>  > the paper does not clarify the relationship between a low drop rate and the selection of the most vulnerable samples.
> We apologize for the unclear text.
>
> **Section 3.1:** We find that a lower drop rate results in increased attack success. However, it requires more partitions, and therefore more models. For example, a drop rate of 5% will lead to (100/5) = 20 partitions. Therefore, for a dataset of size 50,000, each partition will contain 50,000-(50,000*5/100) = 50,000-2,500 = 47,500 points.
>
> **Section 3.2:** Vulnerable point selection means only running the MIA over points that are most vulnerable to leakage, similar to proposed in [5,6,7,8,9]
>
> **Section 4:** We combine the modifications so that we can achieve the same partition size of 47,500 points without having to generate 20 partitions over the entire dataset. First, we identify 5,000 vulnerable points. Instead of creating **20** partitions from the entire dataset, we can create **1** partition (i.e., two subsets of size 5,000/2=2,500 points ) from only the most vulnerable points. This means  50,000-(2,500) = 47,500 points. This effectively means that our partition size is the same as we would have gotten by using a drop rate of 5% over the entire dataset.
>
> In short, we find the 5,000 vulnerable points, split them into **two** subsets and train **one** model on each. Since more models improve attack accuracy, we compare our methodology against a vanilla attack (i.e., random subset of 5,000 points instead of the vulnerable ones) that used **100** models. Results in Table 3 show that our method with two models *outperforms* the vanilla methodology even when using 100 models.
>
> > More experimental details are also needed.
>
> We provide the details in Section 4.1.
>
> Model: Resnet 18
>
> Datasets: cifar10/100, Tiny Imagenet
>
> Training Parameters: 100 epochs, triangular LR 0.4, weight decay 5e 4.
>
> Attack Params: We use the privacy meter code base [2].
>
> If the reviewer believes we are missing any information, please feel free to let us know.
>
>
> > Code is not released for reproducibility.
>
> We use the privacy meter code base [2], as we mentioned in Section 3.1.
>
>
> [1] https://arxiv.org/pdf/2312.03262
>
> [2] https://github.com/privacytrustlab/ml_privacy_meter/tree/d32734161a3395211fe5f3cd461932290b1fafbe
>
> [3]https://github.com/privacytrustlab/ml_privacy_meter/blob/d32734161a3395211fe5f3cd461932290b1fafbe/research/2024_rmia/main.py#L395
>
> [4] https://arxiv.org/pdf/2111.09679
>
> [5] Evaluations of Machine Learning Privacy Defenses are Misleading, Michael Aerni, Jie Zhang, Florian Tramèr, CCS 2024
>
> [6] 1610.05820v2.pdf
>
> [7] 2111.09679v4.pdf
>
> [8] Thomas Steinke and Jonathan Ullman. 2020. The Pitfalls of Average-Case Differential Privacy. DifferentialPrivacy.org. https://differentialprivacy.org/averagecase- dp/.
>
> [9] Membership Inference Attacks from first principles

---

> > ### Author Response · Authors · 2024-11-24
> >
> > > The paper proposes a partitioning strategy based on outlier detection. Does this approach impact membership inference attack results for inliers?
> >
> > No. The two are mutually exclusive. However, inliers are harder to expose with MIAs. So we focus wholly on the vulnerable outliers instead. This provides us a simple and reliable way to identify membership inference with very low FPR.
> >
> > > Does the proposed improvement strategy rely entirely on the accuracy of the chosen outlier detection method, ApB? How should the approach handle data points that are missed or incorrectly identified by the detection method?
> >
> > The strategy relies on using the combination of a low drop rate and the outlier detection method to modify existing MIAs. Since ApB can produce false positives, we do not just select the points with the lower ApB scores as vulnerable to MIA. To guarantee whether or not a point is leaking, we need to execute MIA on it. However, the combination of low drop rate and the outlier detection method reduces the computational complexity required to execute the MIA.
> >
> > >  One motivation for the modified partitioning strategy is to reduce the computational budget, but it remains unclear if the additional cost for outlier detection is acceptable. Quantitative assessments of computational cost and scalability analysis with increasing model complexity would offer practical insights for real-world application.
> >
> > We outline this very assessment in Table 3. Our method is able to achieve strong results while needing to train only **three** models (one model to calculate ApB, and two for the MIA). On the other hand, existing attacks require either training **dozens** [1] or **hundreds** of models [9]. And even then, they are not able to match our success rates (Table 3).

---

> > ### Comment · Reviewer_FBG3 · 2024-11-25
> >
> > Thank you for clarifying some unclear points. I still have some questions:
> >
> > - In the response, the authors primarily argue from the perspective of MIA evaluation metrics to explain why the focus should be on "the most vulnerable samples." However, from an attacker's standpoint, if the method mainly targets a small number of vulnerable samples, how can the trade-off between focusing on these samples and assessing overall privacy leakage be balanced? Providing the MIA performance on non-outlier samples would help clarify this point.
> >
> > - The paper mentions that the choice of outlier detection methods significantly impacts the performance of MIA. Is the outlier detection method robust? For instance, would the effectiveness of the method be affected in scenarios where the target model's data distribution changes or when data augmentation techniques are applied?

---

> > > ### Author Response · Authors · 2024-11-25
> > >
> > > Thank you for engaging with us in the discussion:
> > > 1. Our method can easily be extended for assessing a model's overall privacy. Table 2 shows that the stronger a point's outlier score, the more vulnerable it is to membership inference. The attacker can use this leverage this knowledge to asses overall leakage: 1) Select the top N points, where N<dataset size, 2) partition the N points, 3) train the shadow models, and 4) run the MIA. Up to now, the attacker is doing exactly what we do in Section 4. However, in order to evaluate the privacy of the remaining points, the attacker can now select the *next* N points and repeat steps 1-4. The attacker can continue to do this until they start noticing a significant reduction in attack success. At this point, the attacker has reached the inlier points and may decide not to proceed further. This enables the attacker to use their computational resources efficiently: Spend computation on attacking only the outliers without wasting computation on inliers.
> > >
> > > 2. This is a great point. Outlier methods are not perfect. As a result, in order to *guarantee* whether a point is vulnerable to leakage or not, they need to run the MIA. However, any MIA would suffer from the same issue (i.e., data distribution changes).

---

> > > > ### Author Response · Authors · 2024-11-25
> > > >
> > > > Dear Reviewer FBG3,
> > > >
> > > > Thank you for your valuable feedback. We have provided detailed responses to your questions and hope they address your concerns thoroughly. As the interactive discussion period is coming to a close, please don’t hesitate to reach out if you need further clarification or additional information.

---

### Official Review · Reviewer_xog3 · 2024-10-30

**Soundness:** 1
**Presentation:** 2
**Contribution:** 1
**Rating:** 3
**Confidence:** 5

**Summary:**

This paper identifies two limitations in previous membership inference attacks: excessive data point exclusion during sampling and the evaluation of attack performance across all data points rather than focusing solely on the most vulnerable ones. The author argues that reducing the drop rate during dataset partitioning and applying membership inference attacks only to outliers can significantly improve the attack success rate at low false positive rates.

**Strengths:**

- The paper benchmarks its proposed approach against state-of-the-art methods using various datasets.

**Weaknesses:**

- The paper demonstrates a misunderstanding of membership inference attack objectives and threat models, leading to flawed assumptions and analyses. First, the authors argue that one primary limitation is sampling bias due to excessive data dropping during partitioning. However, this contradicts the approach taken in leading studies (e.g., Zarifzadeh et al., 2024; Carlini et al., 2022), which suggests that data partitioning for reference model training should be unbiased. These works typically recommend a balanced dropout rate of 50%, where each sample $x$ is present in half the reference models (IN models) and excluded from the other half (OUT models). The authors’ observation of increased success rates at lower dropout rates likely arises from impractical assumptions. Since this paper follows the experimental protocols of previous works, where they use the entire dataset for partitioning (Carlini et al., 2022), each example $x$ is included in exactly half the shadow models’ training sets. In this case, **the training sets of individual shadow models and the target model may partially overlap**. This overlap increases as the dropout rate decreases, leading to artificially inflated success rates since shadow models may inadvertently use data closer to the target model’s dataset. In practical applications, such extensive overlap between shadow and target datasets is improbable, especially with larger overlaps, making the paper’s claims less meaningful.

- In the section limitation, the author suggests that limiting MIA to outliers can significantly improve attack success. However, this claim reflects a fundamental misunderstanding of MIA’s objectives. MIA aims to quantify privacy leakage or perform data auditing under regulations (e.g., GDPR) to determine if specific data resides within the original training model. In practice, the data points for MIA evaluations are typically predetermined, meaning we cannot selectively choose only the most vulnerable ones. Although certain data points may indeed be more susceptible to attacks, **MIA’s goal is to examine privacy leakage across all available training data, regardless of vulnerability, to yield an accurate picture of the model’s privacy risk**. Additionally, the paper's claim that recent work advocates evaluating only the most vulnerable points is incorrect. Recent studies emphasize improving evaluation metrics, specifically achieving a low false positive rate (FPR), rather than focusing on selective evaluation targets.

- Given the above analysis, it is unsurprising that this paper’s evaluation results outperform existing baselines, as the methodology is based on unrealistic threat models. The extensive overlap between shadow and target datasets, combined with evaluation focused solely on highly vulnerable points, leads to overly optimistic results that are unlikely to translate to practical scenarios.

- Minor issues. The citation style should be corrected. Use \citet{} when the author(s) names are part of the sentence and \citep{} when the entire citation is parenthetical.

**Questions:**

Please see my previous comments.

**Details Of Ethics Concerns:**

No ethical concerns are involved.

---

> ### Author Response · Authors · 2024-11-22
>
> We thank the reviewer for their feedback. We will address their points below:
>
> >  However, this contradicts the approach taken in leading studies (e.g., Zarifzadeh et al., 2024; Carlini et al., 2022), which suggests that data partitioning for reference model training should be unbiased.
>
> We thank the reviewer for this observation. Zarifzadeh et al., 2024 and Carlini et al., 2022 do *not* refer to bias during **data partitioning**, but during **reference model** selection. Specifically,  Zarifzadeh et al., 2024 state that “we need to make sure the **reference models** are sampled in an unbiased way” [1]. Similarly, Carlini et al state "half of these **models** are trained on the target point (x, y), and half are not" [9]. This means that if we train 10 models without point x, then we need to train exactly 10 models with point x in the dataset. To ensure adherence to this constraint and that there is no reference model bias, we use the same code base for our experiments [2] as the Zarifzadeh et al., 2024 (as we mention in the paper). As a result, we are already adhering to this constraint (Specifically, in lines 395-400 [3]).
>
> > In practical applications, such extensive overlap between shadow and target datasets is improbable,
>
> The reviewer is right to point out that extensive dataset overlap might be impractical in some settings. However, “This is a totally acceptable assumption in the setting of privacy auditing.”[4]. In this scenario, we assume the attacker can sample directly from the target model's training data (e.g., if the model owner wants to audit their model). For this setting, our evaluation shows that sampling bias does impact attack accuracy. We agree with you that this finding might not apply outside privacy auditing, and we would be happy to make this clarification in the paper.
>
>
>
> > MIA’s goal is to examine privacy leakage across all available training data, regardless of vulnerability, to yield an accurate picture of the model’s privacy risk.
>
> We thank the reviewer for this point. However, by studying the most vulnerable points, we are relying on the notion of  *individual privacy*. This notion is accepted within the community and has been echoed by a number of papers [5,6,7,8,9]. The most important of these works is by Aerni et al [5], who state “privacy is not an average-case metric! [54]”. They lay out several reasons for this:
>
> 1. “A blatantly non-private defense that fully leaks one training sample passes existing evaluations.” This demonstrates that even improved evaluation metrics fail to capture the true leakage in a model.
>
> 2. “Existing metrics can be arbitrarily “diluted” by adding new members for which a defense preserves privacy…” giving the illusion of privacy.
>
> 3. “this metric (and prior ones) [i.e., average case metrics] fail to properly capture *individual privacy.*”. Specifically, “If a model violates the privacy of an individual, that individual likely does not care whether the model also leaks 0.1% or 10% of the remaining samples; the individual cares about the fact that an attacker can confidently recover their data.”
>
> 4. Aggregating over the most vulnerable samples (instead of the entire dataset) helps “membership inference attacks to lower-bound the DP guarantees of an algorithm [27, 43, 53, 60].”
>
> These are some reasons why an individual notion of privacy is more important than average-case statistics. In this paper, we are not introducing a new notion, but instead, relying on an existing one that is accepted within the community.
>
> Similarly, pre-selecting points for privacy evaluation (as the reviewer suggests is done for GDPR) does not expose other points that might be vulnerable. Our use of the ApB proxy exposes these other points without having to train dozens of models (as required by using existing MIA). We would be happy to clarify this in the paper.
>
> [1] https://arxiv.org/pdf/2312.03262
>
> [2] https://github.com/privacytrustlab/ml_privacy_meter/tree/d32734161a3395211fe5f3cd461932290b1fafbe
>
> [3]https://github.com/privacytrustlab/ml_privacy_meter/blob/d32734161a3395211fe5f3cd461932290b1fafbe/research/2024_rmia/main.py#L395
>
> [4] https://arxiv.org/pdf/2111.09679
>
> [5] Evaluations of Machine Learning Privacy Defenses are Misleading, Michael Aerni, Jie Zhang, Florian Tramèr, CCS 2024
>
> [6] 1610.05820v2.pdf
>
> [7] 2111.09679v4.pdf
>
> [8] Thomas Steinke and Jonathan Ullman. 2020. The Pitfalls of Average-Case Differential Privacy. DifferentialPrivacy.org. https://differentialprivacy.org/averagecase- dp/.
>
> [9] Membership Inference Attacks from first principles

---

> ### Author Response · Authors · 2024-11-25
>
> Dear Reviewer xog3,
>
> Thank you for your valuable feedback. We have provided detailed responses to your questions and hope they address your concerns thoroughly. As the interactive discussion period is coming to a close, please don’t hesitate to reach out if you need further clarification or additional information.

---

> ### Comment · Reviewer_xog3 · 2024-11-25
>
> Thank you for your response. I have carefully read your response, here are my questions.
>
> First, as acknowledged by the author, there is extensive overlap between the datasets used by the shadow models and the target models. This overlap can lead to artificially inflated success rates since shadow models may inadvertently leverage data that is more representative of the target model’s dataset. This is a straightforward conclusion, as the adversary inherently possesses significantly more knowledge about the target model’s training data, enabling more effective inference. Consequently, this diminishes the first contribution claimed in the paper. Second, while the response asserts that assuming the adversary can directly sample from the target model’s training data is practical in data auditing settings, I agree that this assumption is valid for such specific cases. However, this assumption has limited applicability, particularly because most membership inference studies focus on general attack settings where such knowledge of the target model’s training data is unavailable. As such, it remains unclear how this study advances membership inference attacks (as claimed in the title) beyond these niche scenarios. Third, while I agree that privacy should not be evaluated using average-case metrics, this principle should primarily be reflected in the evaluation strategy rather than the attack method design. Furthermore, it is widely accepted in the community that evaluations should be conducted using randomly sampled datasets to measure the general attack effectiveness. I have listed several references below. During the evaluation, it is standard practice to use a true positive rate (TPR) at a low false positive rate (FPR) as a measure of worst-case leakage instead of average-case leakage. Importantly, this does not involve selectively identifying vulnerable points for evaluation.
>
> I have tried to approach this paper with a supportive perspective; however, there appear to be significant flaws in its scientific contribution and threat model. I would suggest reframing the paper. One potential direction could be to argue that prior state-of-the-art attacks perform poorly on vulnerable points and that the proposed outlier detection method can effectively identify these points, thereby enhancing the overall effectiveness of existing state-of-the-art attacks, which could be demonstrated by the experiment under standard settings.
>
> [1] Carlini, Nicholas, et al. "Membership inference attacks from first principles." 2022 IEEE Symposium on Security and Privacy (SP). IEEE, 2022.
>
> [2] Bertran, Martin, et al. "Scalable membership inference attacks via quantile regression." Advances in Neural Information Processing Systems 36 (2024).
>
> [3] Zarifzadeh, Sajjad, Philippe Liu, and Reza Shokri. "Low-Cost High-Power Membership Inference Attacks." Forty-first International Conference on Machine Learning. 2024.
>
> [4] Li, Hao, “SeqMIA: Sequential-Metric Based Membership Inference Attack.” Proceedings of the 2024 ACM SIGSAC Conference on Computer and Communications Security. 2024.

---

> ### Author Response · Authors · 2024-11-27
>
> We thank the reviewer for engaging with us in this discussion and appreciate your support and perspective. After reading the review and the cited papers again, we agree with your point of view. Specifically, the reviewer has the following main criticisms:
>
> - Attack requires significant overlap between shadow and target,
> - Attack evaluations are not usually over most vulnerable points.
>
> The setting where the above assumptions are true (i.e., access to target data and motivation to identify most vulnerable points) is when a model owner wants to 1) efficiently and 2) effectively quantify the leakage in their model. Current attacks do not meet this criteria because they either require training too many models or are too imprecise. We will promptly modify the paper accordingly.
>
> We have really appreciated this feedback. If there is any other way we can improve our paper, please do not hesitate to let us know.

---

> > ### Comment · Reviewer_xog3 · 2024-12-02
> >
> > Thank you for your response and the revised manuscript. The updated version looks better than the original, at least within the current threat model and problem settings, the technical contributions and experiments make sense.
> >
> > A potential way to better reframe the paper within the context of data auditing is by emphasizing that the privacy guarantees provided by differential privacy mechanisms, such as DP-SGD, are often overly conservative and based on worst-case scenarios. Choosing privacy parameters solely based on theoretical analysis can lead to significant utility loss. This work could make a stronger case by proposing an empirical method to measure the actual privacy guarantees of a model and determine privacy lower bounds through membership inference attacks. It would also be beneficial to include experiments under DP-protected models to substantiate these claims.
> >
> > However, the current scope seems to significantly deviate from that of the original manuscript. Additionally, the proposed approach appears limited in its applicability to broader scenarios. For these reasons, I will adjust my score to 3.

---

> ### Author Response · Authors · 2024-11-30
>
> Dear Reviewer,
>
> Thank you for your feedback. We modified the story of the paper to reflect your suggestions. Specifically, we modified the abstract and intro, and provided a threat model section. We do mention that our threat model is applicable only to auditing scenarios where the model owner wants to identify data leakage. As a result, we assume a large shadow-target overlap. We also mention that this threat model is not applicable to outsider attackers, since they are not likely to have complete knowledge of the target training data.
> As we near the discussion deadline, please let feel free to let us know if you have any more questions or suggestions.

---

### Official Review · Reviewer_6csv · 2024-10-31

**Soundness:** 3
**Presentation:** 3
**Contribution:** 2
**Rating:** 5
**Confidence:** 4

**Summary:**

The paper investigates the problem of improving the evaluated performance (including accuracy and TPR at fixed FPR) of membership inference attacks, via (1) increasing the member/non-member ratio in sampling the training dataset for the target model; and (2) only performing membership inference attack on a subset of selected outlier data records (instead of the whole training dataset). Extensive experiments on CIFAR-10, CIFAR-100, and Tiny-Imagenet datasets show the effectiveness of the proposed method in improving the evaluated performance of MIAs, across various attack strategies.

**Strengths:**

1. Interesting experimental investigations of how to increase the member/non-member ratio and subselect the most vulnerable data records to improve the evaluated performance of membership inference attacks.
2. Novel experimental observations that certain outlier detection methods (such as ApB) could effectively identify data records with high information leakage.

**Weaknesses:**

Under the modified data sampling and selection schemes, the paper lacks discussions regarding why the evaluated MIA performance remains meaningful and comparable with prior works. See questions for details.

**Questions:**

1. Could the authors provide the exact equations for computing the MIA performance (accuracy, TPR, FPR, and AUC) under the modified data sampling/selection schemes? Specifically, how many member/non-member instances are there?

2. Is the TPR in Table 3 computed over the entire training dataset, or the TPR is only averaged over the subset of vulnerable data records selected by the outlier detection method? If it is the former, then a naive strategy that always guesses `member` for non-outlier records and guesses `non-member` for outlier records could achieve a 0.95 TPR at zero FPR (because the selected outlier records consist of 5% of the total data).

3. Could the authors comment on the significance of the evaluated MIA performance under modified data sampling/selection schemes? Given examples as mentioned in question 2, it may not be fair to directly compare the performance under different data sampling/selection schemes. It would help clarify if the authors relate the evaluated performance to the same quantity (e.g., DP lower bound) to understand the fair improvement compared to prior works.

---

> ### Author Response · Authors · 2024-11-23
>
> Under the modified data sampling and selection schemes, the paper lacks discussions regarding why the evaluated MIA performance remains meaningful and comparable with prior works. See questions for details.
> Questions:
> > Could the authors provide the exact equations for computing the MIA performance (accuracy, TPR, FPR, and AUC) under the modified data sampling/selection schemes? Specifically, how many member/non-member instances are there?
>
> As for the specific equations, we use the privacy meter implementations available in [1]. In Table 3, we are running our attacks over 5,000 points. We partition the points into two subsets of 2,500 points each. For the modified attacks, we use the 5,000 most vulnerable points. For the unmodified implementation, we run over 5,000 random points. This was done to ensure a fair comparison. This means 50,000-(5,000/2) = 47,500 points in each shadow model dataset.  As a result, MIA performance statistics are over 5,000 points.
>
> > Is the TPR in Table 3 computed over the entire training dataset, or the TPR is only averaged over the subset of vulnerable data records selected by the outlier detection method? If it is the former, then a naive strategy that always guesses member for non-outlier records and guesses non-member for outlier records could achieve a 0.95 TPR at zero FPR (because the selected outlier records consist of 5% of the total data).
>
> It is the latter, over the most vulnerable examples. The simple strategy, that the reviewer suggested, might work in *some* settings. However, ApB is not perfect and could produce false positives. To account for this case, running the MIA would guarantee whether or not there is privacy leakage.
>
> > Could the authors comment on the significance of the evaluated MIA performance under modified data sampling/selection schemes? Given examples as mentioned in question 2, it may not be fair to directly compare the performance under different data sampling/selection schemes. It would help clarify if the authors relate the evaluated performance to the same quantity (e.g., DP lower bound) to understand the fair improvement compared to prior works.
>
> We thank the reviewer for this comment. Our goal is to improve MIA performance from the notion of individual privacy [2]. Specifically, how much leakage can we *confidently* detect with lowest computational cost. To do so, we focus on data selection and sampling strategies. Therefore, the significance of the MIA performance is due to the ability to correctly infer membership of a significant chunk of data without having to train hundreds of models.
>
> In the context of individual privacy, the model owner wants to identify data most vulnerable to leakage. Therefore, a fair comparison would be to evaluate our changes against what is available to them in current literature.
>
> We are not entirely sure what the reviewer means by "relate the evaluated performance to the same quantity (e.g., DP lower bound)".  Any clarification would be helpful.
>
>
> [1] https://github.com/privacytrustlab/ml_privacy_meter/tree/d32734161a3395211fe5f3cd461932290b1fafbe
> [2] "Evaluations of Machine Learning Privacy Defenses are Misleading," Michael Aerni, Jie Zhang, Florian Tramèr, CCS 2024

---

> ### Author Response · Authors · 2024-11-25
>
> Dear Reviewer 6csv,
>
> Thank you for your valuable feedback. We have provided detailed responses to your questions and hope they address your concerns thoroughly. As the interactive discussion period is coming to a close, please don’t hesitate to reach out if you need further clarification or additional information.

---

> ### Comment · Reviewer_6csv · 2024-11-26
>
> Thanks for the clarifications.
>
> > In Table 3, we are running our attacks over 5,000 points. We partition the points into two subsets of 2,500 points each. For the modified attacks, we use the 5,000 most vulnerable points.
>
> I see, so effectively the member/non-member ratio in evaluation is still $1 : 1$, is this correct? If so, it would help clarify if the authors distinguish the discussion about member/non-member ratio for training target model, versus the member/non-member ratio for evaluating the attack.
>
>
> > We are not entirely sure what the reviewer means by "relate the evaluated performance to the same quantity (e.g., DP lower bound)". Any clarification would be helpful.
>
> An example would be [Theorem 5.2, a], where the membership inference attack advantage is upper bounded under $(\varepsilon, \delta)$-differential privacy, where the target model is trained under a training dataset formed by randomly including each canary data point with probability $\frac{1}{2}$.
>
> In the authors' experiment, there appears to be two modifications: the $\frac{1}{2}$ sampling probability is modified to be a higher value $p$, and the canary data is modified to be outliers detected by ApB. The latter modification is okay, while the first modification would increase the upper bound for MIA advantage under $(\varepsilon, \delta)$-DP, by [Proposition 5.7 a].  Consequently, it is not fair to compare the MIA performance (advantage in this context) under different $p$. One fairer way is to compare the audited lower bound for $\varepsilon$ under different $p$.
>
> **Reference**
>
> [a] Steinke, T., Nasr, M., & Jagielski, M. (2024). Privacy auditing with one (1) training run. Advances in Neural Information Processing Systems, 36.

---

> > ### Author Response · Authors · 2024-11-30
> >
> > >  If so, it would help clarify if the authors distinguish
> >
> > We would be happy to clarify this in the camera ready version of the paper.
> >
> > > while the first modification would increase the upper bound for MIA advantage
> >
> > Thank you for your feedback. That is correct. However, we clarify the story of the paper to show that our goal is to provide a tool the model's owner can find the most vulnerable samples. Since the model owner has complete knowledge of the target, they have the freedom to choose a different value for $p$. At the same time, using canaries does not expose the actual vulnerable points in the dataset (Section 3.2). Our modifications enable the model owner to identify the most vulnerable samples in a 1) reliable and 2) efficient manner so that they can take precautionary action.

---

> > > ### Author Response · Authors · 2024-12-02
> > >
> > > Dear Reviewer,
> > > Thank you for engaging with us. We hope we answered your questions and concerns. Please let us know if you have any other questions or suggestions.

---

### Official Review · Reviewer_6tLi · 2024-11-03

**Soundness:** 3
**Presentation:** 3
**Contribution:** 2
**Rating:** 5
**Confidence:** 4

**Summary:**

This paper addresses an important limitation in Membership Inference Attacks (MIAs), i.e., their true positive rate is extremely low under strict false positive constraints, often nearing random guessing. The authors enhance MIA performance by addressing sampling bias and refining attack aggregation methods, resulting in improved true positive rates (TPR) even at very low false positive rates (FPRs). Their approach achieves near-zero FPR while delivering a near-perfect Area Under the Curve (AUC).

**Strengths:**

1. The paper’s findings on the relationship between dropout rate and true positive rate are intriguing. The discovery that the shadow model should approach the leave-one-out ideal state is meaningful.
2. The experimental results are impressive, which demonstrates a substantial boost in AUC.
3. The paper is well-structured, making complex concepts accessible and easy to follow.

**Weaknesses:**

1. The novelty of the work is unclear. Privacy research on outliers has been extensively explored, as in the paper “Membership Inference Attacks from First Principles”.  If the approach primarily involves filtering these samples, the novelty seems incremental.
2. The description of settings for the shadow model pool and the victim model’s training data is unclear, such as the training data size, pool size, etc. Besides, it would be helpful to include some ablation studies on these settings.
3. It’s debatable of one of this paper’s argument, which believes that evaluating the average attack accuracy over the entire dataset is a flaw. It appears more as an evaluation of certain MIA performance. Thus, the second claimed contribution may just reflect an alternative display of attack results, rather than a true improvement. This raises questions about the paper’s contribution.
4. One of the optimizations is the reduction in partitions, but it’s unclear from the experiments where this is applied or how this benefit can be quantified.
5. Presentation issues: The appendix presentation is somewhat rough, with figures lacking captions and Table 4 missing all borders.

**Questions:**

1. Taking the LiRA method as an example, if the dropout rate is reduced to 10%, would this imply that, for a single data point, it is labeled as membership much more often than non-membership? Could you explain in detail how this method affects the number of positive and negative samples and the impact of these sample counts on true positive and false positive rates?
2. Could the proposed method be limited by a strong dependency between the training dataset sampling size and the pool used for the shadow model? For instance, if the training dataset contains 10,000 samples but the pool has 50,000, would the observation about the relationship between dropout rate and AUC still hold?
3. It’s unclear how the experiments specifically support the claim that your method reduces the number of partitions. Could you provide more explicit experimental evidence demonstrating this advantage?

---

> ### Author Response · Authors · 2024-11-23
>
> > One argument in the paper regarding the evaluation of average attack accuracy over the entire dataset is debatable. It appears to be more of an evaluation of certain Membership Inference Attack (MIA) performances, rather than a fundamental flaw. Thus, the second contribution may simply reflect an alternative way to present attack results, rather than a genuine improvement.
>
> We appreciate this feedback. However, by focusing on the most vulnerable data points (instead of average case statistics), we employing the concept of *individual privacy*, which is an accepted notion in the community and echoed by several papers [5,6,7,8,9]. Notably, Aerni et al. [5] state that "privacy is not an average-case metric!" [54] and provide several reasons:
>
> 1. “A blatantly non-private defense that fully leaks one training sample passes existing evaluations.” This demonstrates that even improved evaluation metrics fail to capture the true leakage in a model.
>
> 2. “Existing metrics can be arbitrarily “diluted” by adding new members for which a defense preserves privacy…” giving the illusion of privacy.
>
> 3. “this metric (and prior ones) [i.e., average case metrics] fail to properly capture *individual privacy.*”. Specifically, “If a model violates the privacy of an individual, that individual likely does not care whether the model also leaks 0.1% or 10% of the remaining samples; the individual cares about the fact that an attacker can confidently recover their data.”
>
> 4. Aggregating over the most vulnerable samples (instead of the entire dataset) helps “membership inference attacks to lower-bound the DP guarantees of an algorithm [27, 43, 53, 60].”
>
> This demonstrates that an individual-centric approach to privacy is more meaningful than average-case statistics. In this paper, we are not introducing a new privacy concept but building on an existing, widely accepted one. We would be happy to clarify this further in the paper.
>
>
> [1] https://arxiv.org/pdf/2312.03262
> [2] https://github.com/privacytrustlab/ml_privacy_meter/tree/d32734161a3395211fe5f3cd461932290b1fafbe
> [3] https://github.com/privacytrustlab/ml_privacy_meter/blob/d32734161a3395211fe5f3cd461932290b1fafbe/research/2024_rmia/main.py#L395
> [4] https://arxiv.org/pdf/2111.09679
> [5] "Evaluations of Machine Learning Privacy Defenses are Misleading," Michael Aerni, Jie Zhang, Florian Tramèr, CCS 2024
> [6] 1610.05820v2.pdf
> [7] 2111.09679v4.pdf
> [8] Thomas Steinke and Jonathan Ullman. 2020. "The Pitfalls of Average-Case Differential Privacy." DifferentialPrivacy.org. https://differentialprivacy.org/averagecase-dp/
> [9] "Membership Inference Attacks from First Principles"

---

> ### Author Response · Authors · 2024-11-23
>
> > The novelty of this work is not entirely clear. Privacy research on outliers, as exemplified in the paper "Membership Inference Attacks from First Principles," has been extensively explored. If the primary approach involves filtering these samples, the contribution appears incremental.
>
> While the cited paper does make significant progress in the right direction, it has notable limitations:
>
> - “their evaluation methodology still computes an attack’s success at identifying membership (i.e., the TPR) across all members.” [5].
> - As a result, their reported TPR at 0% False Positive Rate (FPR) is significantly lower compared to our modified version (see Table 3).
> - Their approach requires training hundreds of models, which is computationally expensive and might not be possible for large models.
>
> > The settings for the shadow model pool and the training data for the victim model are unclear, including details like the training data size and pool size. It would also be helpful to include ablation studies on these settings.
>
> We apologize for the oversight. We evaluated our method on three datasets: CIFAR-10, CIFAR-100, and Tiny ImageNet, with dataset sizes of 50k, 50k, and 100k samples respectively. The pool sizes, as well as the number of samples used to train each individual shadow model, are 47.5k, 45.5k, and 97.5k, respectively (see Section 4.1).
>
> The reviewer also mentioned conducting an ablation study. We already do so in this version of the paper, by first evaluating the two modifications *individually* to measure improvement in attack success (Section 3). Only then do we combine the two in Section 4 to maximize attack improvement.
>
>
>
>
> >Taking the LiRA method as an example, if the dropout rate is reduced to 10%, would this imply that, for a single data point, it is labeled as membership much more often than non-membership?
>
> If the drop rate is 10%, then it does *not* mean that a single point is a member more than it is a non-member, as it would bias reference model selection [1]. We have to ensure that the number of positive and negative samples remains balanced. In other words, we select an equal number of models that have point x in the dataset, and point x outside the dataset. This is ensured by the privacy metric code base we use for evaluation [3]. A 10% drop rate only means that when we train a shadow model, we train it on 90% of the available data. However, this would mean 10 partitions to cover the entire dataset (100%/10% = 10 partitions). Our goal is to overcome this limitation so that we can have a small drop rate while still requiring fewer partitions (Section 4).
>
>
> >Could the proposed method be limited by a strong dependency between the training dataset sampling size and the pool used for the shadow model?
>
> This is a great observation. Yes, training data and pool size does impact AUC [7]. For example, in the *ideal* world, if the training dataset has 10,000 points, the shadow model should be trained on 9,999 points or a leave-one-out scenario. Even though this would result in a powerful attack, it would be too computationally expensive to execute.
>
>
> >It’s unclear how the experiments specifically support the claim that your method reduces the number of partitions. Could you provide more explicit experimental evidence demonstrating this advantage?
>
> We apologize for the unclear text.
>
> **Section 3.1:** We find that a lower drop rate results in increased attack success. However, it requires more partitions, and therefore more models. For example, a drop rate of 5% will lead to (100/5) = 20 partitions. Therefore, for a dataset of size 50,000, each partition will contain 50,000-(50,000*5/100) = 50,000-2,500 = 47,500 points.
>
> **Section 3.2:** Vulnerable point selection means only running the MIA over points that are most vulnerable to leakage, similar to proposed in [5,6,7,8,9]
>
> **Section 4:** We combine the modifications so that we can achieve the same partition size of 47,500 points without having to generate 20 partitions over the entire dataset. First, we identify 5,000 vulnerable points. Instead of creating **20** partitions from the entire dataset, we can create **1** partition (i.e., two subsets of size 5,000/2=2,500 points ) from only the most vulnerable points. This means  50,000-(2,500) = 47,500 points. This effectively means that our partition size is the same as we would have gotten by using a drop rate of 5% over the entire dataset.
>
> In short, we find the 5,000 vulnerable points, split them into **two** subsets and train **one** model on each. Since more models improve attack accuracy, we compare our methodology against a vanilla attack (i.e., random subset of 5,000 points instead of the vulnerable ones) that used **100** models. Results in Table 3 show that our method with two models *outperforms* the vanilla methodology even when using 100 models.

---

> > ### Author Response · Authors · 2024-11-25
> >
> > Dear Reviewer 6tLi,
> >
> > Thank you for your valuable feedback. We have provided detailed responses to your questions and hope they address your concerns thoroughly. As the interactive discussion period is coming to a close, please don’t hesitate to reach out if you need further clarification or additional information.

---

> > > ### Comment · Reviewer_6tLi · 2024-11-26
> > >
> > > Thank you for your rebuttals, which provided additional details about the experimental setups. However, my major concern is that, the improvement in attack AUC seems to stem primarily from changes to the target dataset. For an individual sample, its vulnerability to privacy attacks does not appear to be affected by whether the target dataset is filtered. With that said, I still find it difficult to fully understand and appreciate the novelty of this work. So I'd like to maintain my current score.

---

> > > > ### Author Response · Authors · 2024-11-30
> > > >
> > > > Dear Reviewer,
> > > >
> > > > Thank you for your feedback. We changed the motivation of the paper to show that our modifications help the model owner expose vulnerable samples efficiently and reliably. This is a major weakness of current attacks that do not fully leverage the complete knowledge of the train data and limited compute available to the model owner. While the vulnerability of the individual sample remains unchanged, our modifications help us find these samples in an efficient and reliable way.

---

> > > > > ### Comment · Reviewer_6tLi · 2024-12-03
> > > > >
> > > > > Thank you for your reply. It's meaningful to focus on those more vulnerable samples to emphasize the harm of MIA. However, there are related works from this perspective, such as the paper titled "Understanding Data Importance in Machine Learning Attacks, Does Valuable Data Pose Greater Harm" published in NDSS 2025. Besides, I think the way of exploring this topic by demonstrating the improvement of MI attack is not very straightforward. I hope this work can explain the core idea more clearly, including making comparisons with related works.

---

> > > > > > ### Author Response · Authors · 2024-12-04
> > > > > >
> > > > > > We thank the reviewer for this comment. We would be happy to compare the NDSS findings with our own. However, we want to note that that cited work does not provide an efficient method to identify vulnerable points. In contrast, our goal is to reduce the cost and improve accuracy of the MIA.

---

### Meta-Review · Area_Chair_4bjC · 2024-12-16

**Metareview:**

Thank you for your submission to ICLR!
The reviewers generally noted that the proposed approach is interesting, but were concerned of an unfair apples-to-oranges comparison with prior works.

Reviewers noted a lack of novelty, as the role of outliers had been investigated rather thoroughly in prior works (e.g., see https://arxiv.org/abs/2206.10469). There were also concerns about the presentation of results, which did not make it clear how the different methods were compared and where the improvements stem from.

The global view was thus towards rejection.

**Additional Comments On Reviewer Discussion:**

The rebuttal addressed some of the reviewers' concerns, but the global sentiment remained among all reviewers that the setting and threat model should be clarified, to better understand where the improvements over prior work stem from.

---

### Decision · Program_Chairs · 2025-01-22

Reject